# Erector Spinae Plane Block Decreases Chronic Postoperative Pain Severity in Patients Undergoing Coronary Artery Bypass Grafting

**DOI:** 10.3390/jcm11195949

**Published:** 2022-10-09

**Authors:** Marcin Wiech, Sławomir Żurek, Arkadiusz Kurowicki, Beata Horeczy, Mirosław Czuczwar, Paweł Piwowarczyk, Kazimierz Widenka, Michał Borys

**Affiliations:** 1Second Department of Anesthesia and Intensive Therapy, Medical University of Lublin, Staszica 16, 20-081 Lublin, Poland; 2Department of Cardiac Surgery, Medical Faculty, University of Rzeszow, Lwowska 60, 35-301 Rzeszow, Poland; 3Pro-Familia Hospital, Medical College of Rzeszow University, Witolda 6B, 35-302 Rzeszow, Poland

**Keywords:** chronic postoperative pain, erector spinae plane block, coronary artery bypass grafting, Neuropathic Pain Symptom Inventory

## Abstract

Up to 56% of patients develop chronic postsurgical pain (CPSP) after coronary artery bypass grafting (CABG). CPSP can affect patients’ moods and decrease daily activities. The primary aim of this study was to investigate CPSP severity in patients following off-pump (OP) CABG using the Neuropathic Pain Symptom Inventory (NPSI). This was a prospective cohort study conducted in a cardiac surgery department of a teaching hospital. Patients undergoing OP-CABG were enrolled in an erector spinae plane block (ESPB) group (*n* = 27) or a control (CON) group (*n* = 24). Before the induction of general anesthesia, ESPB was performed on both sides under ultrasound guidance using 0.375% ropivacaine. The secondary outcomes included cumulative oxycodone consumption, acute pain intensity, mechanical ventilation time, hospital length of stay, and postoperative complications. CPSP intensity was lower in the ESPB group than in the CON group 1, 3, and 6 months post-surgery (*p* < 0.001). Significant between-group differences were also observed in other outcomes, including postoperative pain severity, opioid consumption, mechanical ventilation time, and hospital length of stay, in favor of the ESPB group. Preemptive ESPB appears to decrease the risk of CPSP development in patients undergoing OP-CABG. Reduced acute pain severity and shorter mechanical ventilation times and hospital stays should improve patients’ satisfaction and reduce perioperative complications.

## 1. Introduction

Coronary artery bypass grafting (CABG) is one of the common types of cardiac surgeries performed worldwide, with 44 procedures per 100,000 individuals completed annually [1]. Up to 56% of patients develop chronic postsurgical pain (CPSP) after CABG [2,3]. CPSP following CABG surgery can lower patients’ moods and decrease performance of daily activities [3]. A Cochrane meta-analysis found that thoracic epidural analgesia (TEA) could prevent CPSP in patients following thoracic surgery [4]. In our previous study, we showed that continuous paravertebral block lowered CPSP severity and reduced the incidence of CPSP after thoracic surgery [5]. However, not much is known about the use of regional anesthesia techniques in the prevention of CPSP in cardiac surgery. 

Erector spinae plane block (ESPB) is a relatively new regional anesthesia technique described by Forero [6]. In the years since its introduction, ESPB has been used in different types of surgical procedures, including cardiac surgery [7,8,9]. In previous research, we described this type of fascial block in patients undergoing mitral and/or tricuspid valve repair via a right mini-thoracotomy and off-pump CABG (OP-CABG) [10,11]. However, we did not evaluate the incidence of CPSP in the months following patient discharge. This study aimed to assess the severity and incidence of CPSP in patients undergoing OP-CABG via sternotomy with preemptive, bilateral ESPB. 

## 2. Materials and Methods 

This was a prospective cohort study conducted in a cardiac surgery department of a teaching hospital. The study protocol was approved by the Bioethics Committee of the Medical University of Lublin, Lublin, Poland (permit number KE-0254/219/2018). Informed consent was obtained from the patients, and the study was conducted in accordance with the tenets of the Declaration of Helsinki for medical research involving human subjects.

### 2.1. Participants

Patients undergoing an OP-CABG procedure were enrolled in an ESPB group or a control (CON) group. Patients were enrolled consecutively. The same two surgeons performed each surgery. First, we recruited patients to the CON group, then to the ESPB group. The inclusion criteria were adult patients (≥18 years) scheduled for elective surgery. Patients with chronic pain at admission, a history of alcohol or recreational drug abuse, known bleeding disorders, allergies to the drugs used during the study, antidepressant or epileptic drug treatment, and chronic use of painkillers were excluded. 

### 2.2. General Anesthesia

For induction of general anesthesia, the following was used: 0.2–0.4 mg/kg of etomidate, 2–4 µg/kg of fentanyl, and 0.6 mg of rocuronium. Fentanyl infusion was continued at a flow of 25–100 µg/hour, and sevoflurane (0.5–1.0 minimal alveolar concentration) was administered for anesthesia maintenance. The patients received additional doses of rocuronium every 30–40 min and norepinephrine or nitroglycerine as required. The physicians who anesthetized the patient adjusted hemodynamic parameters within 25% of the patient’s baseline using anesthetics and vasopressors.

### 2.3. Regional Block and Postoperative Care

In the ESPB group, before the induction of general anesthesia, single-shot bilateral ESPB was performed under ultrasound guidance, as described in our previous studies [10,11]. On each side, 0.2 mL·kg^−1^ of 0.375% ropivacaine (Ropimol, Molteni, Italy) was administered. The total volume of local anesthetic solution did not exceed 40 mL per patient. 

About 20–30 min before the end of the surgery, the patients received 0.1 mg per kg of oxycodone hydrochloride (up to 10 mg) intravenously (i.v.) and acetaminophen (1.0 g i.v.). Each patient was then transferred to the Intensive Care Unit (ICU). In each case, extubation and weaning from mechanical ventilation were performed according to the attending physician’s discretion. The attending physician measured acute postoperative pain intensity using the Numerical Rating Scale (NRS, 0–10) immediately after extubation. The attending nurse then assessed pain severity every 6 h. 

The mechanical ventilation time was defined as a period from the end of anesthesia to extubation. Each patient was transferred to the Intensive Care Unit (ICU). Then, an attending anesthesiologist assessed the possibility of extubating the patient in the ICU. The ventilatory settings were continuous positive pressure or pressure support ventilation with positive end-expiratory pressure below 7 mbar, pressure support below 3 mbar, and the fraction of inspired oxygen not exceeding 30%. The suggested respiratory parameters for extubation were respiratory rate below 20/min, tidal volume over 6 mL of the patient’s ideal body weight, and oxygen saturation (SpO_2_) > 93%, or similar to the patient’s basal SpO_2_ before preoxygenation. The patient’s heart rate and systolic blood pressure should not have exceeded 25% of the basal measurements obtained before the induction of anesthesia. Moreover, the patient should have responded to questions addressed by a physician. If the patient had not been extubated, an attending physician continued ventilation under sedation with propofol and fentanyl. The second attempt for extubation was done the next day. The patient received a bolus of oxycodone (0.1 mg/kg) when sedation was stopped. Then, the next attempt for extubating was performed. 

Standard pain treatment included oxycodone administered i.v. via a patient-controlled analgesia pump (1 mg/mL, 1 mL bolus, 5 min refraction time). In addition, the patients received 1.0 g of acetaminophen every 6 h, 100 mg of ketoprofen i.v. twice daily, and ondansetron i.v. (4 mg twice daily) as nausea and vomiting prophylaxis. In cases of severe pain (i.e., exceeding 4 on the NRS), the attending nurse was permitted to administer a bolus of oxycodone (5 mg). 

### 2.4. Persistent Postoperative Pain 

For the assessment of persistent postoperative pain, we used the Neuropathic Pain Symptom Inventory (NPSI) developed by Bouhassira et al. [12], as employed in our previous studies [13,14]. All patients were interviewed via telephone 1, 3, and 6 months post-surgery. 

### 2.5. Outcomes 

The primary outcome was CPSP severity 1, 3, and 6 months surgery. The secondary outcomes included the cumulative oxycodone dose, acute pain intensity on the NRS, mechanical ventilation time, hospital length of stay, and postoperative complications. Patients without NPSI results were excluded from further analysis. 

### 2.6. Statistics

The Student’s *t*-test was used to analyze parametric data. The data are presented as means and 95% confidence intervals. Nonparametric data were calculated using the Mann–Whitney *U* test and are presented as medians and interquartile ranges. Categorical variables were analyzed using Fisher’s exact test. Multivariate logistic regression was used to reveal the parameters that affect CPSP occurrence. The odds ratio (OR) was used to describe the predictors that were included in the model. The receiver operating characteristic (ROC) curve was calculated for the best model. All measurements were performed using Statistica 13.1 software (Stat Soft. Inc., Tulsa, OK, USA).

## 3. Results

The study was conducted from March 2019 to April 2020. In total, 74 patients were enrolled. Overall, 23 patients were lost to follow-up. The final study comprised data from 51 patients: 24 in the CON group and 27 in the ESP group (Figure 1). Patient demographics, time of mechanical ventilation, length of hospital stay, and preoperative results of the ASA scoring systems are presented in Table 1. 

### 3.1. Primary Outcome 

CPSP severity was significantly higher in the CON group than in the ESPB group 1, 3, and 6 months after OP-CABG surgery (Table 2). As presented in Table 3, fewer patients in the ESPB group showed signs of persistent pain. 

Table 3 shows the number of patients who experienced CPSP 1, 3, and 6 months post-surgery.

### 3.2. Secondary Outcomes 

Acute pain severity was significantly lower in the ESPB group than in the CON group (Table 4). The patients in the ESPB group used less oxycodone via PCA than the CON group (4 [2–8] vs. 25 [20–25] mg, *p* < 0.001) (Figure 2). Moreover, patients in the CON group received more rescue doses of oxycodone than the ESPB group (5 [0–5] vs. 0 [0–0], *p* = 0.0012). 

The mechanical ventilation time was shorter in the ESPB group than in the CON group (2 [1–3] vs. 10.5 [8–13.25] h, *p* < 0.001). Moreover, the length of hospital stay was shorter in the ESPB group than in the CON group (7 [6–9] vs. 10 [8–12] days, *p* < 0.001). There was no between-group difference in postoperative complications. Figure 1 presents postoperative oxycodone consumption administered via PCA.

We found three variables associated with CPSP occurrence at month 6. The best model was found using stepwise logistic regression. The area under the ROC curve was 0.899 for this model. Two variables positively associated with CPSP occurrence were additional doses of oxycodone and the mechanical ventilation time, which were positively associated with OR 1.65 (1.164–2.339, *p* = 0.005) and 1.303 (1.053–1.614, *p* = 0.015). The anesthesia time was negatively associated with CPSP incidence at month 6 (0.989 (0.982–0.996), *p* = 0.002). 

## 4. Discussion

Our results suggest that preemptive ESPB can alleviate CPSP severity months after an OP-CABG procedure (Table 2). As shown in Table 3, fewer patients had signs of persistent pain after the regional block. Moreover, acute pain severity and oxycodone consumption were minor in patients following the regional block. The regional block group also had shorter postoperative mechanical ventilation and hospital length of stay times. The results of the logistic regression could suggest that CPSP occurrence at month 6 was associated with incidences of breakthrough pain in the postoperative period, which required additional doses of opioids. 

We showed in the present study that bilateral ESPB could reduce CPSP severity and incidence. According to the results of previous research, regional anesthesia procedures might prevent the occurrence of CPSP [4,15]. The effectiveness of TEA in reducing the incidence of CPSP in patients following thoracic surgery was presented in a study by Lu et al. [16]. Kairaluoma et al. showed a reduction of CPSP intensity by preemptive paravertebral block in breast surgery patients [17]. The authors of this trial performed a single preemptive injection of a local anesthetic. We reported a reduction in the incidence and intensity of CPSP in patients following thoracotomy [5]. However, in this study, continuous paravertebral block was used. In our recent study, we presented that quadratus lumborum block can reduce CPSP severity in patients undergoing nephrectomy [18]. However, the effect of new regional anesthesia techniques on preventing CPSP development requires further studies. 

Limited treatment modalities can prevent CPSP development following cardiac surgery procedures [19]. In a meta-analysis by Carley et al. concerning prophylaxis of CPSP with drugs following surgeries, only gabapentoids were showed to prevent persistent pain three months after cardiac surgery [20]. However, as stated by the authors of this meta-analysis, the results should be interpreted cautiously due to the small study sizes.

New regional anesthesia techniques have been explored extensively in acute postoperative pain treatment [21,22]. We confirmed in the present study that ESPB could reduce acute pain intensity and opioid demands in patients undergoing OP-CABG. In a trial by Krishna et al., bilateral ESPB reduced pain intensity in patients undergoing cardiac surgery procedures requiring cardiopulmonary bypass [7]. Nagaraja and colleagues showed that ESPB was comparable to continuous TEA in the reduction of acute pain severity in patients following a cardiac surgery procedure [8]. Macaire et al. presented that continuous ESPB improved patient postoperative rehabilitation after open cardiac surgery and reduced opioid consumption [23]. 

Our study has some limitations. It was an observational study. Thus, selection bias is possible. In addition, we enrolled a relatively small group of patients, and we did not examine the quality of regional block with the pinprick technique. Patients in the CON group did not receive a sham block with saline. There was no blinding in our study. 

To conclude, our results suggest that preemptive ESPB can decrease the risk of CPSP development in patients after an OP-CABG procedure. The lower acute pain severity and shorter mechanical ventilation times and hospital stays associated with the procedure should improve patients’ satisfaction and reduce hospital costs and perioperative complications. 

## Figures and Tables

**Figure 1 jcm-11-05949-f001:**
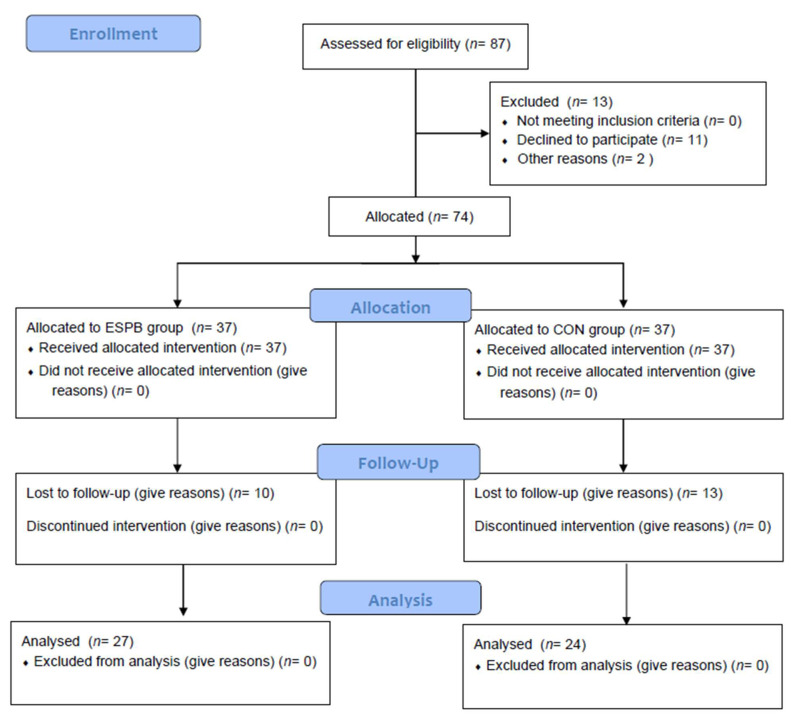
Study flow diagram. ESPB, erector spinae plane block group; CON, control group.

**Figure 2 jcm-11-05949-f002:**
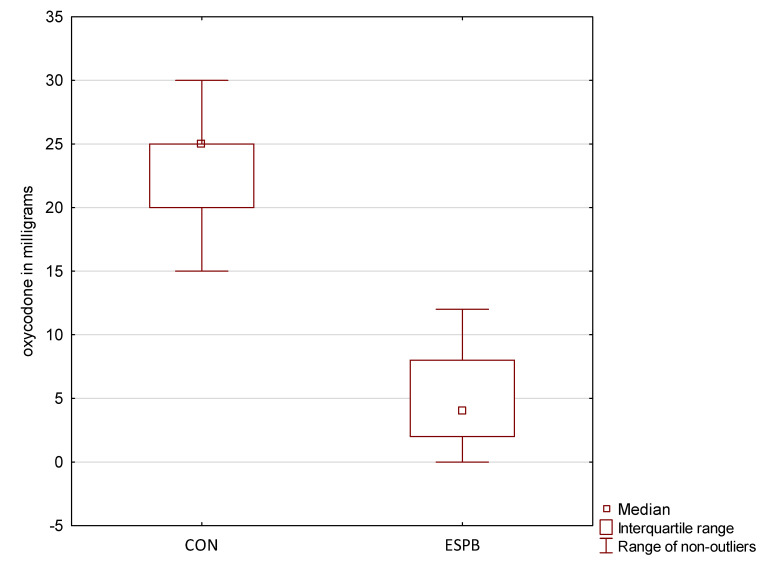
Oxycodone consumption. ESPB, erector spinae plane block group; CON, control group.

**Table 1 jcm-11-05949-t001:** Patient demographics.

Parameters	ESPB	CON	*p* Value
Number of patients	27	24	
Male (%)	24 (88.9)	22 (91.7)	1.0
Age (years)	64.7 (62.0–67.4)	67.1 (63.4–70.7)	0.28
Weight (kg)	85.2 (80.4–90.0)	83.7 (77.9–89.4)	0.68
Height (cm)	171.1 (167.7–174.5)	170.6 (167.1–174.1)	0.85
BMI	29.1 (27.8–30.3)	28.6 (27.1–30.1)	0.65
ASA	3 (2–3)	3 (3–3)	0.45
Anesthesia time (minutes)	194 (174–215)	201 (141–185)	0.63
Surgery time (minutes)	159 (140–179)	163 (177–225)	0.79
Intraoperative fentanyl (mcg)	381 (359–404)	499 (463–535)	<0.001

Patient age, weight, height, BMI, surgery and anesthesia times, and intraoperative fentanyl are presented as means and confidence intervals. ASA is presented as a median and interquartile range. ASA, American Society of Anesthesiologists; BMI, body mass index; ESPB, erector spinae plane block group; CON, control group.

**Table 2 jcm-11-05949-t002:** Severity of persistent postoperative pain.

After Discharge	ESPB	CON	*p* Value
1 month	1 (0–2) *	4 (3–6)	<0.001
3 months	0 (0–1) *	4 (2–6)	<0.001
6 months	0 (0–0) *	2 (14)	<0.001

The severity of post-thoracotomy pain syndrome was detected with the NPSI (0–100). Data are shown as medians (interquartile ranges). * Denotes a significant between-group difference. ESPB, erector spinae plane block group; CON, control group.

**Table 3 jcm-11-05949-t003:** Incidence of chronic postsurgical pain (CPSP).

After Discharge	Number of Patients (%)		*p* Value
	ESPB	CON	
1 month	17 (63) *	23 (96)	<0.01
3 months	8 (30) *	23 (96)	<0.001
6 months	6 (22) *	22 (92)	<0.001

* Denotes a significant between-group difference. ESPB, erector spinae plane block group; CON, control group.

**Table 4 jcm-11-05949-t004:** Acute pain severity.

Hours after Extubation	ESPB	CON	*p* Value
0	3.0 (2.0−4.5) *	5.0 (4.0−5.8)	<0.001
6	2.5 (2.0−4.0) *	4.5 (3.5–5.0)	<0.001
12	3.0 (2.0−3.5) *	4.0 (3.0−4.0)	<0.01
18	2.0 (0.0−3.0) *	3.0 (2.0−4.0)	<0.01
24	0.0 (0.0−1.5) *	3.0 (1.0−3.5)	<0.001

Data are shown as medians (interquartile ranges). * Denotes a significant between-group difference. ESPB, erector spinae plane block group; CON, control group.

## Data Availability

The data presented in this study are available on request from the corresponding author.

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
