# Peer review of "Erector Spinae Plane Block Decreases Chronic Postoperative Pain Severity in Patients Undergoing Coronary Artery Bypass Grafting"

_jcm, 2022, doi:10.3390/jcm11195949_

Round 1
Reviewer 1 Report
Response to jcm-1929628
This study is about the effect of ESPB on the chronic postoperative pain severity in patients undergoing OP-CABG. The analgesic effect of the regional analgesia including ESPB in the cardiac surgery has been investigated in some studies, furthermore, this study explores the intensity of chronic pain after surgery; this is considered as relatively new information presented.
Unfortunately, reporting about details about the method and definition of variables was not sufficient in the report. For instance, no data is provided about the intraoperative opioid consumption, resue analgesic consumption, and the incidence of breakthrough pain, and no detailed definition of mechanical ventilation time. Also, some results seemed unrealistic, so detailed explanations of these issues should be addressed.
See below for a detailed description.
(1) What is the definition of the ‘mechanical ventilation time’?
In my understanding, all patients were transferred to PACU, and extubation was performed there, but it was found that the time difference was more than 8 hours (2 vs. 10.5) just by administering the ESPB intervention than control group. It seemed not general to stay in PACU for an average of 10 hours instead of ICU.
A more detailed explanation is needed on this part.
(2) It was described that pain evaluation was performed immediately after extubation. Was the mental status suitable for pain evaluation at that time?
(3) The variables including ‘Intraoperative opioid consumption, resue analgesic consumption, and the incidence of breakthrough pain’ could be helpful to understand this data. In addition the hemodynamic data during surgery between groups may be helpful.
(4) The incidences of CPSP in the control group in the control group were more than 90%. If this results were true, it was not appropriate management was performed in the control group despite of the use of oxycodone, paracetamol, etc.
It looks more like the result of relative superiority due to the malpractice of the control group rather than the effect of ESPB.
(5) Sample size calculation have not been performed?
(6) Line 80, in oxycodone dose, the ‘per kg’ might be omitted.
(7) The expressions of probability in Table 1, and p values in other tables are mixed.
Author Response
Reviewer 1
This study is about the effect of ESPB on the chronic postoperative pain severity in patients undergoing OP-CABG. The analgesic effect of the regional analgesia including ESPB in the cardiac surgery has been investigated in some studies, furthermore, this study explores the intensity of chronic pain after surgery; this is considered as relatively new information presented.
Unfortunately, reporting about details about the method and definition of variables was not sufficient in the report. For instance, no data is provided about the intraoperative opioid consumption, resue analgesic consumption, and the incidence of breakthrough pain, and no detailed definition of mechanical ventilation time. Also, some results seemed unrealistic, so detailed explanations of these issues should be addressed.
Our response: Thank you for your critical point of view. Some methods could have been described more precisely. We added more details to the main text to present our approach. Some methods were not shown in detail because we focused on CPSP.
(1) What is the definition of the 'mechanical ventilation time'?
In my understanding, all patients were transferred to PACU, and extubation was performed there, but it was found that the time difference was more than 8 hours (2 vs. 10.5) just by administering the ESPB intervention than control group. It seemed not general to stay in PACU for an average of 10 hours instead of ICU.
A more detailed explanation is needed on this part.
Our response: Yes, the reviewer is right. Such a difference in the extubation time cannot be explained by ESPB. In the case of prolonged mechanical ventilation, the patient was sedated with propofol and fentanyl. Each patient was transferred to ICU not PACU. We corrected it in the manuscript. We explained in the main text how and when the patient was extubated after the surgery.
"The mechanical ventilation time was defined as a period from the end of anesthesia to extubation. Each patient was transferred to the Intensive Care Unit (ICU). Then, an attending anesthesiologist assessed the possibility of extubating the patient. The ventilatory settings were continuous positive pressure or pressure support ventilation with positive end-expiratory pressure below 7 mbar, pressure support below 3 mbar, and the fraction of inspired oxygen not exceeding 30%. The suggested respiratory parameters for extubation were respiratory rate below 20 / minute, tidal volume over 6 mL of the patient's ideal body weight, and oxygen saturation (SpO2)> 93%, or similar to the patient's basal SpO2 before the preoxygenation. The patient's heart rate and systolic blood pressure should not have exceeded 25 % of the basal measurements obtained before the induction of anesthesia. Moreover, the patient should have responded to questions addressed by a physician. If the patient had not been extubated, an attending physician continued ventilation under sedation with propofol and fentanyl. The second attempt for extubation was made the other day. The patient received a bolus of oxycodone (0.1 mg/KG) when sedation was stopped. Then, the next attempt for extubating was performed"
Each participant in this study received bolus oxycodone at the end of the surgery. However, an attending anaesthesiologist made the final decision for extubating, which was a subjective one. Due to lower postoperative pain, patients in the ESPB group were extubated shortly after the surgery. Patients in the standard care group did not fulfill the criteria for extubating mentioned above. The second attempt to wean from mechanical ventilation was made the other day.
(2) It was described that pain evaluation was performed immediately after extubation. Was the mental status suitable for pain evaluation at that time?
Our response: Before extubation, as we described above, an attending anaesthesiologist assessed the patient's responsiveness.
(3) The variables including 'Intraoperative opioid consumption, resue analgesic consumption, and the incidence of breakthrough pain' could be helpful to understand this data. In addition the hemodynamic data during surgery between groups may be helpful.
Our response: In Table 1, we inserted a row concerning intraoperative fentanyl doses. We added results concerning oxycodone rescue doses to the results section. An attending nurse assessed the incidences of breakthrough pain. Then, according to a nurse's discretion, an additional dose of oxycodone was given. Thus, we did not record breakthrough pain incidences but extra doses of opioids. We did not record hemodynamic measurements from each anesthesia chart. The physicians anesthetized the patient and adjusted hemodynamic parameters within 25% of the patient's baseline using anesthetics and vasopressors. We added this piece of information to the manuscript.
(4) The incidences of CPSP in the control group in the control group were more than 90%. If this results were true, it was not appropriate management was performed in the control group despite of the use of oxycodone, paracetamol, etc.
It looks more like the result of relative superiority due to the malpractice of the control group rather than the effect of ESPB.
Our response: Each patient received multimodal postoperative analgesia. However, some patients in the control group have moderate to severe pain ( mean NRS 5 after extubation). CPSP incidences and severity were not associated with NRS severity. We added a new analysis to the manuscript, multivariate logistic regression. The result of this analysis showed that the risk of CPSP was positively correlated with additional doses of oxycodone (a surrogate of breakthrough pain incidences), the postoperative ventilation time (longer in the control group), and negatively with anesthesia time ( a little longer in ESPB group). We also mentioned these results in the discussion. "The results of logistic regression could suggest that the incidences of breakthrough pain in the postoperative period which required additional doses of opioids were associated with CPSP incidences at month 6." Thank you for your remarks. We believe that it was not malpractice and each patient in our study received much means to treat postoperative pain, but patients in ESPB group got more.
(5) Sample size calculation have not been performed?
Our response: No, it was not. It was an observational study. This trial can be used as data to calculate the sample size.
(6) Line 80, in oxycodone dose, the 'per kg' might be omitted.
Our response: We added per kg
(7) The expressions of probability in Table 1, and p values inother tables are mixed.
Our response: We unified them as "p value".
Reviewer 2 Report
The authors present the use of an erector spinae plane block (ESPB) to reduce postoperative chronic pain in off-pump coronary artery bypass patients. The authors' result support the use of the ESPB secondary to the reduction of chronic pain when compared to controls. The manuscript is overall well written.
With respect to the placement of the block, however, the authors need to elaborate more. Line 75-76 refers to placement of the block as done in a previous study (ref 12). This reference is to TAP and quadratus lumborum blocks. The authors do not mention if ultrasound was used for placement. This should be included for clarity if employed.
Author Response
Reviewer 2
The authors present the use of an erector spinae plane block(ESPB) to reduce postoperative chronic pain in off-pump coronary artery bypass patients. The authors' result support the use of the ESPB secondary to the reduction of chronic pain when compared to controls. The manuscript is overall well written.
Our response:
Thank you for your remarks.
With respect to the placement of the block, however, the authors need to elaborate more. Line 75-76 refers to placement of the block as done in a previous study (ref 12). This reference is to TAP and quadratus lumborum blocks. The authors do not mention if ultrasound was used for placement. This should be included for clarity if employed.
Our response: Thank you for this remark. We added a piece of information that this block was performed under US guidance. This reference's placement was also incorrect. We fixed the reference order.